# High Prevalence of Hepatitis B Virus Drug Resistance Mutations to Lamivudine among People with HIV/HBV Coinfection in Rural and Peri-Urban Communities in Botswana

**DOI:** 10.3390/v16040592

**Published:** 2024-04-11

**Authors:** Bonolo B. Phinius, Motswedi Anderson, Irene Gobe, Margaret Mokomane, Wonderful T. Choga, Basetsana Phakedi, Tsholofelo Ratsoma, Gorata Mpebe, Joseph Makhema, Roger Shapiro, Shahin Lockman, Rosemary Musonda, Sikhulile Moyo, Simani Gaseitsiwe

**Affiliations:** 1Botswana Harvard Health Partnership, Private Bag BO320, Gaborone, Botswana; bphinius@mail.bhp.org.bw (B.B.P.); manderson@bhp.org.bw (M.A.); gmpebe@bhp.org.bw (G.M.); jmakhema@bhp.org.bw (J.M.); rshapiro999@gmail.com (R.S.); shahin.lockman@gmail.com (S.L.); rmusonda@gmail.com (R.M.); smoyo@bhp.org.bw (S.M.); 2School of Allied Health Professions, Faculty of Health Sciences, University of Botswana, Private Bag UB 0022, Gaborone, Botswana; gobei@ub.ac.bw (I.G.); mokomanem@ub.ac.bw (M.M.); 3Department of Immunology and Infectious Diseases, Harvard T. H. Chan School of Public Health, Boston, MA 02115, USA; 4Division of Medical Virology, Faculty of Medicine and Health Sciences, Stellenbosch University, Stellenbosch, Private Bag X1, Matieland 7602, South Africa; 5School of Health Systems and Public Health, University of Pretoria, Private Bag X20, Pretoria 0028, South Africa

**Keywords:** hepatitis B virus, drug resistance, people living with HIV, Botswana, Africa

## Abstract

(1) Background: We aimed to determine the prevalence of hepatitis B virus (HBV) resistance-associated mutations (RAMs) in people with HBV and human immunodeficiency virus (HBV/HIV) in Botswana. (2) Methods: We sequenced HBV deoxyribonucleic acid (DNA) from participants with HBV/HIV from the Botswana Combination Prevention Project study (2013–2018) using the Oxford Nanopore GridION platform. Consensus sequences were analyzed for genotypic and mutational profiles. (3) Results: Overall, 98 HBV sequences had evaluable reverse transcriptase region coverage. The median participant age was 43 years (IQR: 37, 49) and 66/98 (67.4%) were female. Most participants, i.e., 86/98 (87.8%) had suppressed HIV viral load (VL). HBV RAMs were identified in 61/98 (62.2%) participants. Most RAMs were in positions 204 (60.3%), 180 (50.5%), and 173 (33.3%), mostly associated with lamivudine resistance. The triple mutations rtM204V/L180M/V173L were the most predominant (17/61 [27.9%]). Most participants (96.7%) with RAMs were on antiretroviral therapy for a median duration of 7.5 years (IQR: 4.8, 10.5). Approximately 27.9% (17/61) of participants with RAMs had undetectable HBV VL, 50.8% (31/61) had VL < 2000 IU/mL, and 13/61 (21.3%) had VL ≥ 2000 IU/mL. (4) Conclusions: The high prevalence of lamivudine RAMs discourages the use of ART regimens with 3TC as the only HBV-active drug in people with HIV/HBV.

## 1. Introduction

Hepatitis B virus (HBV) remains a global health concern even in the era of potent vaccines and antiretroviral therapy (ART) that can greatly reduce morbidity and mortality. HBV global prevalence is 3.8%, and HBV infection causes 820,000 deaths annually [1]. Of 296 million people living with chronic hepatitis B (CHB) globally, 82 million are in Africa, second only to the Western Pacific, where 116 million people with CHB reside [1]. Globally, 2.73 million people are coinfected with HIV/HBV, of which 1.96 million reside in sub-Saharan Africa (71%) [2]. HBV has a 3.2 kb genome and it replicates via a reverse transcriptase enzyme that lacks a proof-reading mechanism [3]. Full-length genome analyses of HBV have revealed varying nucleotide differences between different viral strains, hence the classification of genotypes for a difference greater than 7.5% and subgenotypes where nucleotide divergence is between 4–7.5% [3]. Therefore, 9 HBV genotypes denoted A-I, a 10th putative genotype J, and over 35 subgenotypes exist [3]. In Botswana, genotypes A1, D2, D3, and E are reported to be circulating in the country [4,5,6]. Due to its error-prone reverse transcriptase replication, HBV has a mutation rate of 10^−4^ to 10^−6^ substitutions/site/year [7,8,9]. It is through this high mutation rate that allows the virus to evolve to adapt to the host environment and to withstand immune and drug pressure.

There is a call to eliminate HBV by the year 2030, with specific targets to reduce HBV incidence by 95% and mortality by 65% [1]. The use of nucleos(t)ide analogues (NAs) can greatly contribute to these targets, but the development of resistance-associated mutations (RAMs) remains one of the challenges that hinder HBV elimination. Prolonged antiviral (ARV) use without adequate monitoring may lead to the selection of variants with RAMs that reduce ART susceptibility [10]. In the most recent World Health Organization (WHO) CHB treatment guidelines released in March 2024, the WHO recommends treatment of CHB patients who have evidence of significant fibrosis, an HBV viral load of greater than 2000 IU/mL with elevated alanine transaminase (ALT) levels, or presence of co-infections such as HIV [11]. NAs with a high barrier to drug resistance are highly recommended for treatment. These include entecavir (ETV) or tenofovir disoproxil fumarate (TDF) or tenofovir alafenamide (TAF) for infants, adolescents, and adults aged 2 years or older [11]. In individuals with HBV/HIV coinfection, tenofovir (TFV) + lamivudine (3TC) or emtricitabine (FTC) is recommended. Although other drugs such as 3TC and telbivudine are active against HBV, they have a low barrier to resistance and therefore are not recommended for HBV treatment [12,13]. TFV is considered the most effective drug against HBV; however, there is emerging evidence of possible amino acid substitutions that may reduce susceptibility to TFV, such as rtS78T, rtA194T, and rtN236T [14,15,16,17].

Botswana has an HIV prevalence of 20.8% in the adult population [18]. Hepatitis B surface (HBsAg) prevalence differs in different populations studied in Botswana. In infants aged 25 months and below (2016–2018), HBsAg prevalence was 0.67% [19], 1.1% in blood donors (2014–2015) [6], 2.1% in pregnant women (2010–2012) [5], and 9.3% in treatment-naïve people living with HIV (PLWH) (2009–2012) [20]. A recent study from our group recently reported a prevalence of 8% in primarily treatment-experienced and HIV virally suppressed PLWH [21]. Occult hepatitis B infections (OBI) are not usually reported in national HBV prevalence estimates. OBI is defined as the presence of replicative competent HBV deoxyribonucleic acid (DNA) in the blood and liver of HBsAg-negative individuals [22]. OBI prevalence was 6.6% in pregnant women, 26.5% in ART-naïve PLWH, and 33% in primarily ART-experienced PLWH from the above studies [5,21,23].

Botswana has a robust HIV treatment program that has seen the country surpass the United Nations Programme on HIV/AIDS (UNAIDS) 95-95-95 targets at 95-98-98 [18]. Therefore, in Botswana, 95% of PLWH know their HIV status, 98% of those who know their status are on ART, and finally, 98% of those on ART are virally suppressed. In 2016, Botswana adopted the HIV Treat-all program with the first-line regimen being Truvada and dolutegravir [24]. The prior regimen was TFV/FTC/ Efavirenz (EFV) as the first line, while the earliest first-line regimen had 3TC as the only HBV active drug. Current HIV guidelines in Botswana released October 2023 recommend screening for HBV in PLWH who are initiating ART, and those testing positive for HBV are to be maintained on a TDF or TAF regimen whilst those that test HBV-negative will be illegible for Dolutegravir, (DTG)+3TC, or DTG+FTC dual-therapy treatment simplification. The widespread use of ARVs for HIV treatment in the country has improved patient health outcomes among PLWH; however, HBV status, viral load, and RAMs are not monitored in this setting. We therefore aimed to determine the prevalence of HBV RAMs in people with concomitant HBV/HIV in Botswana. The data generated from this project will guide treatment guidelines for PLWH living with HBV coinfection in Botswana.

## 2. Materials and Methods

### 2.1. Study Population

Plasma samples from PLWH recruited in the Botswana Combination Prevention Project (BCPP) (2013–2018) were used in this study. Details of BCPP are described elsewhere [25]. In brief, the BCPP study was a cluster-randomized trial conducted in 15 paired communities matched by size, pre-existing health services, population, age structure, and geographic location. The study enrolled 12,610 participants, which was a random sample of 20% of households in each community. BCPP aimed to assess whether a combination of prevention strategies would reduce HIV incidence at a population level. At baseline, 3596 were PLWH, while 9014 were not living with HIV [25]. Participants signed written informed consent in the BCPP study. Our study was approved by the Human Research Development Committee at the Botswana Ministry of Health (HPDME 13/18/1) with a waiver of consent.

### 2.2. HBV Screening

Participant plasma samples from PLWH previously screened for various HBV serological markers [21] were used. HBsAg screening was performed using the Murex Version 2 HBsAg enzyme-linked immunosorbent assay (ELISA) with a lower limit of detection of 0.13 IU/mL according to the manufacturer’s data (Diasorin, Dartford, UK). In parallel, plasma samples were also screened for total core antibodies (anti-HBc) using the Monolisa anti-HBc PLUS ELISA kit (Bio-Rad, Marnes-la-Coquette, France). Samples with positive HBsAg serology (HBsAg+) were further screened for hepatitis e antigen (HBeAg) and anti-HBc immunoglobulin m (IgM) using the Monolisa HBe Ag/Ab and Monolisa anti-HBc Plus 1 Plaque (Bio-Rad, Marnes-la-Coquette, France), respectively. HBV DNA quantification for HBsAg+ samples and screening for OBI in HBsAg-negative samples was conducted using the COBAS AmpliPrep/COBAS TaqMan HBV Test version 2.0 (Roche Diagnostics, Mannheim, Germany) following the manufacturer’s instructions [21]. This assay has a limit of detection of 20 IU/mL.

### 2.3. Nanopore Sequencing

The QIAamp DNA Blood Mini kit (Qiagen, Hilden, Germany) was used to extract DNA from 200 μL of HBsAg+ and OBI-positive (OBI+) plasma samples according to the manufacturer’s instructions, and we changed the final elution volume to 30 μL. DNA concentration and quality were determined using the Qubit fluorometer (Thermo Fisher Scientific, Waltham, MA, USA). Library preparation was adopted from an already established protocol [26] and modified as previously described [27]. In brief, a two-step polymerase chain reaction (PCR) was performed to amplify the whole HBV genome. Forward and reverse tiling primers [26] were pooled into two primer pools of 10 μm concentration each. Master mixes were prepared for each primer pool, and two PCR assays for each of the primer pool were performed to amplify the whole HBV genome. The first-round master mixes were composed of 1.5 μL of nuclease-free water, 0.05 μL of primer pool, 6.25 μL of Q5^®^ Hot Start High-Fidelity 2X master mix (New England Biolabs, Whitby, ON, Canada), and 5 μL of template volume. PCR conditions were an initial denaturation at 98 °C for 30 s, 35 cycles of a denaturation step at 98 °C for 15 s, annealing and extension steps at 65 °C for 5 min, and finally, a hold step at 4 °C for ∞. Second-round PCR master mixes for each primer pool were composed of 0.5 μL of the primer pool, 3.75 μL of nuclease-free water, 6.25 μL of the Q5^®^ Hot Start High-Fidelity 2X master mix, and 2.5 μL of the template. First-round PCR conditions were used for second-round PCR. The library was quantified using the Qubit fluorometer and loaded into version R9.4.1 flow cells (Oxford Nanopore Technologies, Oxford, UK). HBV sequences were generated using the GridION sequencing platform (Oxford Nanopore Technologies, Oxford, UK).

### 2.4. Sequencing Analysis

Raw FASTQ files were exported and subsequently processed using Guppy, employing dual-indexed reads for base calling and demultiplexing. FASTQ files were uploaded into Genome Detective (version 2.64) for reference assembly. Generated consensus HBV sequences were viewed, aligned and trimmed using AliView alignment viewer [28]. Geno2pheno version 2.0 (https://hbv.geno2pheno.org) (accessed on 11 December 2023) and the Stanford HBVseq version 9.5.1 (https://hivdb.stanford.edu/HBV/HBVseq/development/HBVseq.html) (accessed on 11 December 2023) online databases were used to determine HBV genotypes and RAMs. Furthermore, genotypes were confirmed using phylogenetic analysis. We constructed a maximum-likelihood tree (ML-tree) using the best-fitting model of nucleotide substitution [TVM+F+I+G4] using IQ-TREE with 1000 bootstrap replicates [29,30]. For mutational analysis, 98 HBV sequences had evaluable reverse transcriptase (RT) region coverage. We trimmed the RT position to only have amino acid positions rt1-250, and these amino acid RT sequences were analyzed for HBV RAMs using the Sandford HBVseq tool version 9.5.1 (https://hivdb.stanford.edu/HBV/HBVseq/development/HBVseq.html) (accessed 11 December 2023).

### 2.5. Statistical Analysis

Participants’ sociodemographic and clinical characteristics were summarized in proportions and medians with interquartile ranges (IQR). Categorical data were analyzed using Fishers’ exact test or the Chi-squared test where appropriate, while continuous variables were compared using the Wilcoxon rank–sum test. We used a proportion test to compare the prevalence of RAMs between participants of varying HBV viral load categories. All statistical analyses were conducted using Stata version 18.0 (StataCorp LLC, College Station, TX, USA), and *p*-values less than 0.05 were deemed statistically significant.

## 3. Results

### 3.1. Participant Description

Participant plasma samples positive for HBsAg and OBI were used. From 271 HBsAg+ samples, we quantified HBV viral load in 128 that had sufficient volume. We attempted to sequence all the 128, and 79/128 (61.7%) had a valuable RT region. Out of the 126 OBI+ samples, only 72 samples had sufficient sample volume for DNA extraction and subsequent amplification and sequencing. Out of the 72, only 19/72 (26.4%) had a valuable RT region, hence a much lower sequencing success rate for samples with OBI. Therefore, we had a total of 98 RT sequences for downstream analysis (Figure 1).

The majority of our study participants were female 66/98 (67.3%) and the median age was 43 years (IQR: 37–49). Most participants had suppressed HIV viral load (VL) (86/98, 87.8%) and were on ART (93/98, 94.9%), and most were on a TDF-containing regimen (40/67, 59.7%). The median time on ART was 7.4 years (IQR: 4.5–10.3). Most participants had an HBV viral load of <2000 IU/mL, 58/98 (59.2%), and we also generated HBV sequences from participants whose viral load was target-not-detected (TND) from the assay we used, being 20/98 (20.4%). RAMs were identified in participants with low HBV DNA and those with OBI (Table 1). Participants that had positive HBeAg serology were 13/76 (17.1%), and those with positive anti-HBc IgM were 4/75 (5.3%). The majority of the sequences assessed for RAMs were from participants who resided in the central (40/98) and northern (41/98) part of Botswana. There was no statistically significant difference between participants with and without RAMs in all variables except for HBV viral load categories. Through a proportion test, we observed significantly more RAMs in participants with no detectable HBV viral load compared to those with a viral load of <2000 IU/mL [17/20 (85.0%) vs. 31/58 (53.5%), *p* = 0.01], Table 1.

### 3.2. Resistance Associated Mutations

Overall, subgenotypes A1 (93.9%), D3 (3.1%), and E (3.1%) were identified among the 98 participants. A similar distribution was observed among participants with RAMs (*n* = 61): A1 (90.2%), D3 (4.9%), and E (4.9%). Phylogenetic analysis based on maximum likelihood with 1000 bootstrap replicates was used to construct a tree. Some of the sequences with RAMs were clustered closely together (supported by posterior probability >0.90), suggesting a possible transmission (see Figure 2).

In this study, we identified RAMs which confer resistance to 3TC in 61/98 (62.2%) participants. There were no RAMs identified which conferred resistance to tenofovir in our study; therefore, among participants on TDF-containing regimen, none had HBV TDF RAMs. However, there was one participant who had an amino acid substitution (N236K) at a position that has been reported to confer resistance against tenofovir, rt236. Amino acid substitutions were identified at 7 RT positions, with most being at position rt204 (60.3%), rt180 (50.5%), rt173 (33.3%), and rt250 (6.9%), (see Figure 3a,b). We observed already characterized amino acid substitutions, rtM204V, rtL180M, and rtV173L, which were the most prevalent at 58.9%, 50.5% and 32.3%, respectively. All other mutations had a prevalence below 7%. Other amino acid substitutions were observed in positions known for resistance-associated mutations; however, they had not been well characterized, as shown in Figure 3b.

Uncharacterized mutations mostly appeared in one participant, as shown in Table 2. Some participants [17/61 (27.9%) and 10/61 (16.4%)] had triple mutations (rtV173L, rtL180M and rtM204V) and double mutations (rtL180M and M204V). Of participants with triple mutations, 5/12 (41.7%) were on the 3TC regimen, and among those with double mutations, 5/8 (62.5%) were on a regimen with a 3TC-only backbone, as shown in Table 2. Some of the participants only had a single mutation. These mostly harbored the rtM204V, 8/61 (13.1%), followed by those who only had an rtL180M mutation, 5/61 (8.2%), and those that had the rtV173L mutation, 2/61 (3.3%). Three participants harbored quadruple mutations with different combinations, these being the rtV173L/rtL180M/rtM204V/rtM250L combination, the rtL80V/rtV173L/rtL180M/rtM204I combination, and the rtL80V/rtV173L/rtL180M/rtM204V combination. The latter two quadruple mutations differed only in terms of the amino acid substitution in the same M204 position. In one participant, methionine is substituted by isoleucine, while in another, it is substituted by valine, as shown in Table 2.

## 4. Discussion

Botswana has a robust and successful HIV treatment program which has resulted in the country achieving UNAIDS 95-95-95 goals [18]. However, HBV screening before ART initiation has not been robust, and there is also no monitoring of HBV response to ART once people with HBV/HIV are initiated on ART for HIV. Recent Botswana HIV treatment guidelines emphasize screening of HBV prior to ART initiation for an improved management of people with HBV/HIV. The widespread and prolonged use of ART for HIV without HBV screening and monitoring has potentially led to the selection of drug-resistant HBV variants. In this study, we report the prevalence of HBV RAMs in participants with HBV/HIV from rural and peri-urban communities in Botswana. We generated HBV sequences from samples with varying HBV viral loads, even those deemed to have undetectable HBV viral load by the assay used in the study. This is not common, as HBV sequencing is generally performed for high-viral-load samples, thus limiting the identification of the full spectrum of mutations in a population [14].

The HBV sequences generated in this study were mostly subgenotype A1 (93.9%), which is considerably different from other previous studies in the country. In blood donors, subgenotype A1 prevalence was 36.1% [6], 45.5% in pregnant women [5], and 48% in ART-naïve PLWH [4]. In our current study, our sample size is larger, and sequences are generated from diverse communities across Botswana, as opposed to previous studies that have focused mainly on the capital city Gaborone and its surrounding areas. However, our data are not different from regional data that show that subgenotype A1 is predominant in other southern African countries, such as Zimbabwe, Zambia, and South Africa [31,32,33]. This subgenotype has been associated with increased risk of developing hepatocellular carcinoma compared to non-A genotypes [34] and low-HBV DNA in carriers [35].

We detected RAMs in participants with varying HBV viral load, some with a TND result from the assay we used (which has also been reported elsewhere [36,37]). There was a significantly high prevalence of RAMs in participants with an “undetectable HBV viral load”, which may be a limitation of the assay we used (which could not detect very low HBV viral loads). Therefore, HBV drug resistance is not to only be associated with treatment failure, as it also occurs at low viraemia. Most participants in this study were on TDF-containing or 3TC-containing regimens, which reflects the first-line ART regimens at the time the BCPP study was conducted, as well as historic first-line ART regimens. Most participants had RAMs associated with 3TC therapy. Botswana adopted the Treat-All strategy in 2016 with DTG-based ART as the first-line regimen. However, earlier first-line regimens included 3TC, and many participants in our study had been on 3TC-containing regimens, especially zidovudine/3TC plus EFV or NVP. Considering our study participants’ time on ART, it is not surprising that they would harbor HBV variants with mutations associated with 3TC resistance.

3TC is a low-genetic-barrier drug for HBV [38], hence we identified a high prevalence of 3TC-associated resistance in this study as expected. M204V (58.9%), L180M (50.5%), and V173L (32.3%) were the predominant mutations, and they are known to confer resistance to 3TC and enhance viral replication [39]. A similar pattern has been observed in Gabon [40], and similarly to our study, the predominant RAM was M204V/I (according to a systematic review of HBV RAMS in Africa [17] and globally [41]). The M204V mutation, as shown in our results, occurred either alone or in combination with other mutations, most commonly with L180M and V173L, which are described as compensatory mutations to M204V as they enhance viral replication [40,42]. M204V, which is highly prevalent in our study, is known to confer resistance to both 3TC and ETV. In one study, lamivudine-resistant HBV M204V mutants had an indication of replication capacity under ETV therapy, with an inhibitory concentration of 90 (IC_90_) [43]. RAMs can confer resistance to several antiviral drugs (cross-resistance), which is a cause for concern when exploring treatment options where drug resistance has been established.

Some of the study participants were on a TDF-containing regimen, and we report no RAM conferring resistance to TDF. HBV resistance to TFV is still controversial. A recent review suggests that resistance to TFV may require more than one resistance mutation that confers resistance to other NAs: L180M, A181V/T, M204I/V, and N236T [14]. A study in the United States showed that participants who had been exposed to 3TC took a longer time to achieve HBV viral suppression while on TFV compared to 3TC-naïve participants [44]. Therefore, it is likely that 3TC-associated mutations may eventually lead to decreased susceptibility to TFV. Some of the study participants with 3TC-associated mutations were on a TDF-containing regimen; this could be because they started on 3TC-based ART before switching to a TDF-based regimen. It could also be due to the transmission of variants with 3TC resistance mutations [45]. We report an uncharacterized polymorphism N236K in a position where it may contribute, together with other mutations, resistance to TFV.

## 5. Conclusions

In a population of ART-experienced individuals with concomitant HBV/HIV, the prevalence of HBV RAMs was high, particularly those known to confer resistance to 3TC. The high prevalence of 3TC RAMs in this population discourages the use of ART regimens with 3TC as the only HBV-active drug in people living with HIV/HBV. The presence of HBV RAMs hinders HBV elimination efforts, hence the need to monitor HBV viral loads in people who are living with HBV infection who are on ART; for example, those who continually fail to suppress the HBV viral load can be tested for HBV drug resistance mutations. TDF-associated resistance mutations were not observed while most participants were on TDF, thus supporting the effectiveness of TDF in treating HBV.

## Figures and Tables

**Figure 1 viruses-16-00592-f001:**
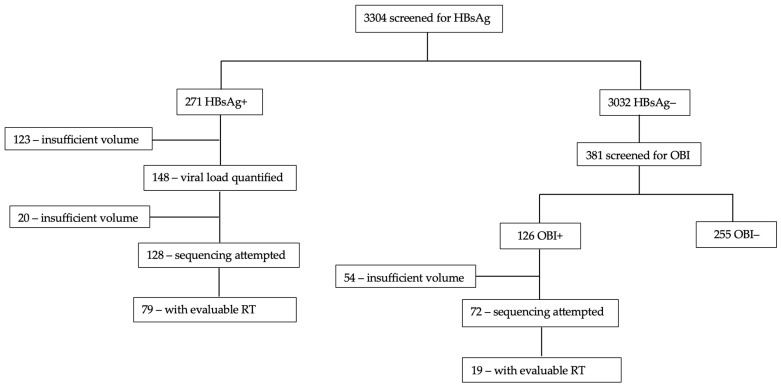
Laboratory flow diagram.

**Figure 2 viruses-16-00592-f002:**
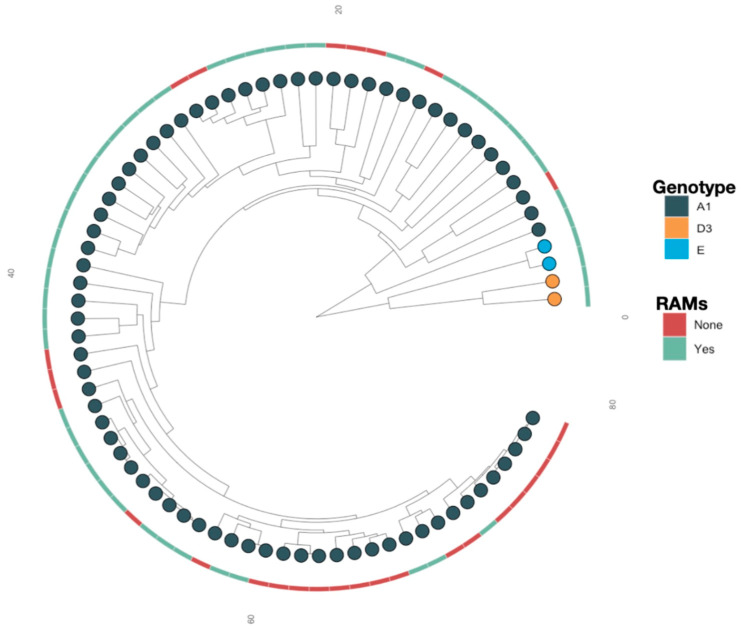
ML phylogenetic tree of representative sequences with and without RAMs by HBV subgenotype.

**Figure 3 viruses-16-00592-f003:**
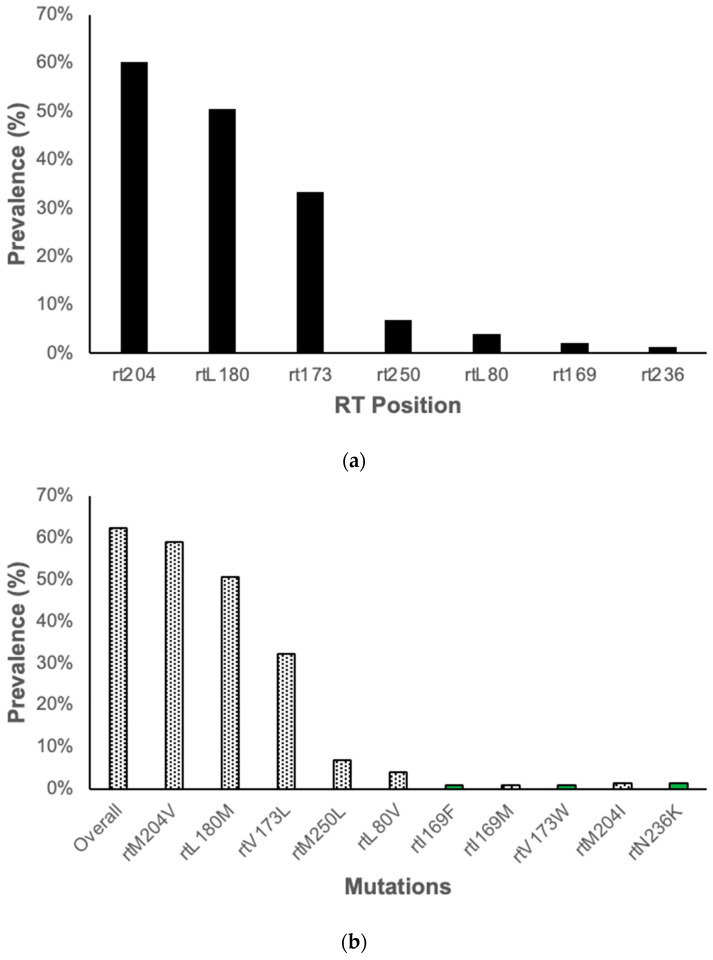
Prevalence of amino acid substitutions ((**a**) by RT position, and (**b**) by specific mutations; black bars indicate mutations that have been characterized, and green bars indicate uncharacterized amino acid substitutions).

**Table 1 viruses-16-00592-t001:** Participant demographic and clinical characteristics.

	No RAMs, *n* = 36	RAMs Present, *n* = 61	*p*-Value
Sex, *n* (%)			
Female	24 (64.9)	42 (68.9)	
Male	13 (35.1)	19 (31.2)	0.82
Age, years, median (IQR)	41 (34–47)	45 (40–50)	0.68
Region			
South	7 (18.9)	10 (16.4)	
Central	17 (46.0)	23 (37.7)	
North	13 (35.1)	28 (45.9)	0.56
Nadir CD4 T cell count, cells/μL, *n* (%), *n* = 19			
<350	7 (70.0)	3 (33.3)	
≥350	3 (30.0)	6 (66.7)	0.11
HIV viral load, copies/mL, *n* (%)			
Suppressed	30 (81.1)	56 (91.8)	
Unsuppressed	7 (18.9)	5 (8.2)	0.12
HBV infection phase, *n* (%)			
HBsAg positive	28 (75.7)	51 (83.6)	
OBI positive	9 (24.3)	10 (16.4)	0.43
HBV viral load, *n* (%)			
TND	3 (8.1)	17 (27.9)	
<2000	27 (73.0)	31 (50.8)	
>2000	7 (18.9)	13 (21.3)	0.04
HBeAg status, *n* (%), *n* = 76			
Negative	24 (88.9)	39 (79.6)	
Positive	3 (11.1)	10 (20.4)	0.30
Anti-HBc IgM, *n* (%), *n* = 75			
Negative	25 (92.6)	46 (95.8)	
Positive	2 (7.4)	2 (4.2)	0.55
Total anti-HBc, *n* (%) *n* = 95			
Negative	7 (19.4)	9 (15.3)	
Positive	29 (80.6)	61 (85.9)	0.44
ART status, *n* (%)			
ART-naive	3 (8.1)	2 (3.3)	
On ART	34 (91.9)	59 (96.7)	0.29
Current ART regimen, *n* (%) *n* = 67			
3TC containing, without TDF	8 (38.1)	19 (41.3)	
TDF containing *	13 (61.9)	27 (58.7)	1.00
Time on ART, years, median (IQR), *n* = 74	5.9 (4.2–10.3)	7.5 (4.8–10.5)	0.55

**Notes:** * all but one participant were on a TDF/FTC-containing regimen; the participant did not have RAMs. **Abbreviations:** RAMs, resistance-associated mutations; HBV, hepatitis B virus; TND, target not detectable; HIV, human immunodeficiency virus; ART, antiretroviral therapy; 3TC, lamivudine; TDF, tenofovir disoproxil fumarate; IQR, interquartile range; HBsAg, hepatitis B surface antigen; OBI, occult hepatitis B infection; HBeAg, hepatitis B e antigen; Anti-HBc IgM, hepatitis B core immunoglobulin M antibodies; anti-HBc, hepatitis B core antibodies.

**Table 2 viruses-16-00592-t002:** Combination of mutations in participants with RAMs.

Mutations	Frequency	ART Status	ART Regimen *^#^
rtV173L/rtL180M/rtM204V	17	All on ART	7 on TDF 5 on 3TC
rtL180M/rtM204V	10	All on ART	3 on TDF 5 on 3TC
rtM204V	8	All on ART	7 on TDF 1 on 3TC
rtL180M	5	All on ART	2 on TDF 2 on 3TC
rtV173L/rtL180M	5	All on ART	2 on TDF 2 on 3TC
rtL180M/rtM204V/rtM250L	3	1 on ART 2 ART naïve	No data
rtV173L/rtM204V	2	1 on ART 1 ART naïve	On 3TC
rtV173L	2	1 on ART 1 ART naïve	No data
rtI169F/rtV173L/rtL180M	1	On ART	On TDF
rtV173L/rtL180M/rtM204V/rtM250L	1	On ART	On TDF
rtL180M/rtM204V/rtN236K	1	On ART	On 3TC
rtL80V/rtV173L/rtL180M/rtM204I	1	On ART	On 3TC
rtL80V/rtV173L/rtL180M	1	On ART	On 3TC
rtL80V/rtV173L/rtL180M/rtM204V	1	On ART	On TDF
rtV173W/rtL180M	1	On ART	On TDF
rtI169M/rtL180M	1	On ART	On TDF
rtM204V/rtM250L	1	On ART	On TDF

**Notes:** * All participants on a TDF-containing regimen also had FTC on the same regimen; ^#^ participants with ART regimen data. **Abbreviations:** ART, antiretroviral therapy; 3TC, lamivudine; TDF, tenofovir disoproxil fumarate; RT, reverse transcriptase.

## Data Availability

The data presented in this study are available upon request from the corresponding author. The data are not publicly available as the sequences are currently being analyzed for other objectives within a bigger project.

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
