# Peer review of "High Prevalence of Hepatitis B Virus Drug Resistance Mutations to Lamivudine among People with HIV/HBV Coinfection in Rural and Peri-Urban Communities in Botswana"

_viruses, 2024, doi:10.3390/v16040592_

Round 1

Reviewer 1 Report

Comments and Suggestions for Authors

General comment

The selection of drug-resistant HBV mutants by antiviral therapy of HBV/HIV coinfection is an important topic which was studied thoroughly by the authors in a region which was heavily affected by this confection. The very large patient cohort was tested with timely methods of virus serology and genome characterization. The paper is well written and the presentation of the data is clear.

Some minor points may be improved as pointed out below.

Specific points

1.      L26. We sequenced HBV DNA …

2.      L71. … Programme on HIV/AIDS (UNAIDS) 95-95-95 targets at 95-98-98 [10]. Please explain the meaning of the 3 numbers.

3.      L73 Explain abbrev. EFV

4.      L92-96. Please provide the limits of detection for HBsAg, HBV DNA (TND) and level of HBV DNA required for sequencing.

5.      Table 1 Please edit: “HBV infection phase” instead of “HBV type”; persistent HBsAg instead of “HBsAg”

6.      L201-205. Three participants had quadruple mutations. Why are they not mentioned?

7.      L261. This sentence is a too strong statement: “We report an uncharacterized polymorphism N236K in a position known to confer resistance to TFV.” I suggest: “… in a position which may contribute among other mutations to resistance to TFV.”

8.      Discussion. Although the study did not use entecavir, the mutations at rt204M described here affect strongly the efficacy of the well-established drug entecavir. Therefore, the discussion may mention the work of Geipel et al. Entecavir allows an unexpectedly high residual replication of HBV mutants resistant to lamivudine. Antivir Ther. 2015;20(8):779-87. doi: 10.3851/IMP2928. PMID: 25560463l

Reviewer 2 Report

Comments and Suggestions for Authors

In current manuscript the retrospective study of presence of drug resistance mutations in reverse transcriptase of HBV in people co-infected with HBV/HIV and receiving the treatment were performed. Broad analysis included 98 patients. The genotype, phylogeny, presence of drug resistance mutations and other parameters were studied. The study demonstrated high prevalence of drug resistance mutations to lamivudine in people under antiretroviral therapy with lamivudine as the only HBV-active drug.

There are only some minor comments:

Line 30 - please clarify the total number of participants 85/97 or 85/98

Line 33 – please, correct the sentence “The triple mutations, rtM204V/L180M/V173L and was the most predominant (17/61 [27.9%]).”

Line 66 – specify if the indicated range of HBV prevalence is given for different regions of Botswana or for different years or for different studies

Line 73 – the EFV abbreviation is given for the first time, please decode it

Line 78 – correct if the HIV, not HBV is mentioned

Line 116 and below - the use of the term consensus in this case - may cause misunderstanding. In this case, the nucleotide sequence of the genome of an individual virus was obtained. The term consensus is more commonly understood as a theoretical representative nucleotide in which each is the one which occurs most frequently at that site in the different sequences which occur in nature.

Figure 3 – it is recommended to remove the decimal points at the graph.
